# High-frequency, year-round time series of the carbonate chemistry in a high-Arctic fjord (Svalbard)

Jean-Pierre Gattuso[1, 2], Samir Alliouane[1], and Philipp Fischer[3]

[1]Sorbonne Université, CNRS, Laboratoire d'Océanographie de Villefranche, 181 chemin du Lazaret, F-06230 Villefranche-sur-mer, France
[2]Institute for Sustainable Development and International Relations, Sciences Po, 27 rue Saint Guillaume, F-75007 Paris, France
[3]Alfred-Wegener-Institut Helmholtz Centre for Polar and Marine Research, Kurpromenade 211, 27498 Helgoland, Germany

**Correspondence:** Jean-Pierre Gattuso (jean-pierre.gattuso@imev-mer.fr)

**Abstract.** The Arctic Ocean is subject to high rates of ocean warming and acidification, with critical implications for marine organisms as well as ecosystems and the services they provide. Carbonate system data in the Arctic realm are spotty in space and time and, until recently, there was no time-series station measuring the carbonate chemistry at high frequency in this region, particularly in coastal waters. We report here on the first high-frequency (1 h), multi-year (5 years) dataset of salinity, temperature, $CO_2$ partial pressure (p$CO_2$) and pH at a coastal site (bottom depth of 12 m) in a high-Arctic fjord (Kongsfjorden, Svalbard). Discrete measurements of dissolved inorganic carbon and total alkalinity were also performed. We show that (1) the choice of formulations for calculating the dissociation constants of the carbonic acid remains unsettled for polar waters, (2) the water column is generally somewhat stratified despite the shallow depth, (3) the saturation state of calcium carbonate is subject to large seasonal changes but never reaches undersaturation ($\Omega_a$ ranges between 1.4 and 3.0) and (4) p$CO_2$ is lower than atmospheric $CO_2$ at all seasons, making this site a sink for atmospheric $CO_2$ (-9 to -16.8 mol $CO_2$ m$^{-2}$ yr$^{-1}$, depending on the parameterisation of the gas transfert velocity). Data are available on PANGAEA: https://doi.pangaea.de/10.1594/PANGAEA.957028.

## 1 Introduction

Despite their major importance, Arctic shelves are among the coastal areas which are understood the least. The Arctic Ocean only covers 4.3% of the total ocean area but has a continental shelf considerably larger than other oceans (52.7% of its total area vs less than 18% globally; Jakobsson et al., 2004; Menard and Smith, 1966) and the total length of its coastline affected by the presence of permafrost represents around 34% of the world coastline (Lantuit et al., 2012). It contains less than 1% of ocean water but receives 11% of the global runoff (Shiklomanov, 1998) and is responsible for 7-10% of the global burial of organic carbon (Stein and Macdonald, 2004).

The Arctic region is one of the "reasons for concern" of the Intergovernmental Panel on Climate Change (IPCC; O'Neill et al., 2017). The Arctic Ocean exhibits the fastest and largest changes which already have impacts on the biota and biogeochemical cycles (Wassmann et al., 2010). The increase in sea surface temperature over the last two decades is similar to, or only slightly higher than, the global average (Fox-Kemper et al., 2021) . However, the greatest future warming is in the Arctic

Ocean, where multi-model mean warming in 2080–2099 can exceed 2 to 5 °C relative to 1995–2014, depending on the $CO_2$ emissions scenario considered (Kwiatkowski et al., 2020).

The massive release of anthropogenic $CO_2$ also generates ocean acidification, a process that describes the increase in dissolved inorganic carbon and bicarbonate and the decline of pH and the saturation state of calcium carbonate minerals. The decrease in pH is projected to be larger in the surface Arctic Ocean than elsewhere, with model mean declines that can exceed 0.45 pH units in SSP5-8.5 (2080–2099 anomalies relative to 1995–2014)(Kwiatkowski et al., 2020).

Freshwater input via rivers and glacier melting have a profound impact on the seawater carbonate chemistry. It decreases total
alkalinity, the seawater buffering capacity and the calcium carbonate saturation state (Fransson et al., 2015). Undersaturation of surface water with respect to aragonite-type $CaCO_3$ was first reported for 2008 in the Canada Basin, preceding other open ocean basins (Zhang et al., 2020). Much of Arctic shallow waters are undersaturated with respect to calcium carbonate, especially aragonite. This is due to the decrease of salinity resulting from increased river runoff and sea ice melt in the summer (Chierici and Fransson, 2009), and to the degradation of organic matter in runoff waters and shelf areas (e.g., Anderson et al., 2017).
Aragonite undersaturation has consequences on aragonite-shelled organisms such as pteropods (e.g., Comeau et al., 2011).

The remoteness and harsh environmental conditions make it difficult to gather carbonate chemistry data in the Arctic, although some coastal sites are easily accessible year round. The goal of this paper is to provide the first high-frequency, multi-year dataset of salinity, temperature, dissolved inorganic carbon, total alkalinity, $pCO_2$ and pH.

## 2    Material and methods

Data were collected at the COSYNA/MOSES-AWIPEV underwater observatory operated since 2012 in Kongsfjorden, an Arctic fjord located on the west coast of Spitsbergen (Svalbard) at 78°55'50.37" N and 11°55'12.10" E (Fischer et al., 2017) (Fig. 1). The study site is coastal (11 m depth $\pm$ 0.7 m of tidal amplitude) and is relatively sheltered in the inner part of the Kongsfjorden, with average tidal currents of 0.1 m $yr^{-1}$. Kongsfjorden is a typical Arctic fjord with minimum winter water temperatures of -1.9 to 0.8 °C in February and March, and maximum average water temperatures of more than 6 °C in August
(see Appendix A). Until 2006, the fjord was regularly covered by sea ice in winter (Gerland and Renner, 2007). Before 2006, the sea ice typically extended into the central part of the fjord, but during the last decade the sea-ice extent has often been reduced to the northern part of the inner bay (Pavlova et al., 2019).

### 2.1    The COSYNA/MOSES-AWIPEV observatory

The COSYNA/MOSES-AWIPEV underwater observatory comprises a land-based FerryBox system (Fig. 1a) equipped with a
set of sensors (Table 1). The FerryBox receives water from 11 m depth from an underwater pump (Fig. 1b and c). To prevent biofouling of the sensors, every night at 00:10, a sulfuric acid (4% for 10 min) flush of the entire sensor system was followed by a rince with freshwater (30 min) prior to switching again to measuring mode. Data were not used for a total duration of 60 min after the initiation of the flush. The observatory also comprises a profiling sensor carrier (REMOS) fitted with another set of sensors that can be remotely-controlled (Fig. 1d and Table 1). The profiling unit is positioned, for varying durations (median:

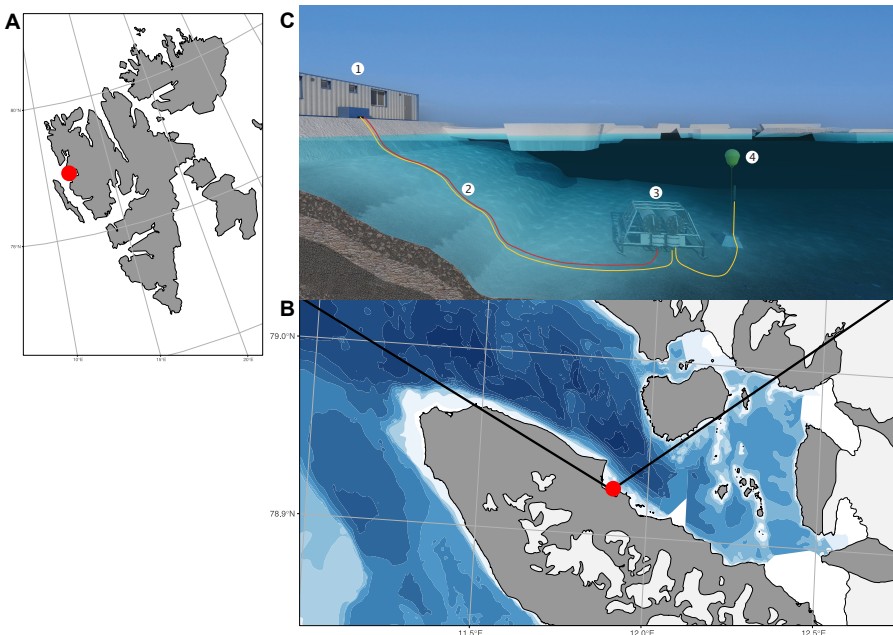

**Figure 1.** Svalbard (A), Kongsfjorden and Ny-Ålesund (B), and observational set-up ()C). 1: FerryBox system, 2: underwater cable and underwater tubes for water supply for FerryBox system, 3: underwater node with water pumps and 4: underwater profiling sensor carrier unit (REMOS). The maps were produced by the R package ggOceanMaps (Vihtakari, 2022).

6 h), in one of the following distances from the sea bottom 1, 3, 5, 7 or 9 m. The effective water depth of the system changed with the tide cycle for at most 1.5 m, but the system itself had a fixed position above ground. For a more detailed description of the Svalbard underwater observatory see Fischer et al. (2017) and Fischer et al. (2020).

The salinity (conductivity) sensor in the FerryBox had some failures. The gaps were filled by salinity values measured with the *in situ* CTD when the REMOS was below 8 m. Such gap filling was not performed for temperature which warms by about 1°C before reaching the FerryBox.

## 2.2 Discrete sampling and measurements

Seawater was sampled in the FerryBox, at about weekly frequency. It was collected into duplicate 500 ml borosilicate glass bottles after a careful rinse. Samples were immediately poisoned with mercuric chloride as described by Dickson et al. (2007). Dissolved inorganic carbon ($C_{\mathrm{T}}$) and total alkalinity ($A_{\mathrm{T}}$) were analysed within 6 months via potentiometric titration following methods described by Edmond (1970) and DOE (1994), by Service National d'Analyse des Paramètres Océaniques du CO₂ at Sorbonne University, France. The average accuracy of $C_{\mathrm{T}}$ and $A_{\mathrm{T}}$ measurements was 2.6 and 3 $\mu$mol kg$^{-1}$ , respectively, compared to seawater certified reference material (CRM) provided by A. Dickson (Scripps Institution of Oceanography). The following CRM batches were used: 148, 155, 165,173, 182 et 196. Repeatability of replicate samples was better than 3 $\mu$mol kg$^{-1}$.

**Table 1.** Sensors deployed in the FerryBox and profiling system. All sensors in the FerryBox system are maintained once a year and all sensors of the profiling system are changed once a year and sent to the manufacturer for maintenance and calibration. The salinity sensors were calibrated according to the standard Unesco procedure (IOC et al., 2010).

| Location | Parameters and sensors | Year of installation |
| --- | --- | --- |
| FerryBox | Water temperature ($^{\circ}$C), SeaBird SBE45 | 2012 |
| | Conductivity (ms m$^{-1}$) / Salinity, SeaBird, SBE45 | 2012 |
| | Oxygen (%), Aanderaa 4175C | 2012 |
| | Chl-a (mg m$^{-3}$), Seapoint Chlorophyll Fluorometer | 2012 |
| | Turbidity (FTU), SeaPoint turbidity meter | 2012 |
| | Partial pressure of $CO_2$ ($\mu$atm), Kongsberg Maritime, HydroC CO2 FT | 2015 |
| REMOS profiling system | Pressure (dbar), Sea&Sun CTD90 | 2012 |
| | Water temperature ($^{\circ}$C), SBE 38 Digital Oceanographic Thermometer | 2015 |
| | Conductivity (ms m$^{-1}$) / Salinity, Sea&Sun CTD90 - ADM 7 pole electrode cell | 2012 |
| | Oxygen (%), Sea&Sun CTD90 - Aanderaa 4175C | 2012 |
| | Chl-a (mg m$^{-3}$), Sea&Sun CTD90 - Cyclops7 Fluorometer | 2012 |
| | Turbidity (FTU), Sea&Sun CTD90 - Seapoint turbidity meter | 2012 |
| | Photosynthetically available radiation (SeaBird), ECO-PAR | 2015 |
| | pH (total scale), SeaBird SeaFET | 2017 |

Unless flagged as of poor quality, $C_T$ and $A_T$ of replicate bottle samples were averaged. When the difference between duplicates was larger than 10 $\mu$mol kg$^{-1}$, the replicate closer to the general trend was kept and the other discarded. The number of outliers discarded was 38 and 41, respectively for $C_T$ and $A_T$ (out of a total number of samples of 229 and 236).

Starting in November 2018, seawater was sampled at approximately monthly interval for pH measurements both in the FerryBox and in the field, below 8 m with a Niskin bottle, to calibrate the pH sensors. Samples were preserved as described by Dickson et al. (2007). pH was measured spectrophotometrically within 6 months of sampling as described in Dickson et al. (2007) using purified m-cresol purple (purchased from Robert H. Byrne's laboratory, University of South Florida). Three to four replicate measurements were performed for each sample on a Cary 60 UV-Vis spectrophotometer (Agilent Technologies). Repeatability was very good: the standard deviation of the replicates ranged from 0.00033 to 0.0091 pH units and the average of 44 mean standard deviations was 0.002 pH units.

## 2.3 Partial pressure of $CO_2$

The measuring range of the HydroC CO2 FT sensor (Contros Kongsberg Maritime) is 200-1000 $\mu$atm, resolution is $< 1$ $\mu$atm and accuracy is $\pm$ 1% reading. The sensor was positioned first in the loop of sensors of the FerryBox in order to avoid alteration of pCO$_2$ through exposure to air. Two sensors were swapped every year and while one was monitoring pCO$_2$, the other one

was factory-calibrated. pCO₂ was measured continuously and data logged every minute. Calibration of the unit was performed by the supplier. It comprised a post-deployment calibration (to assess the drift), a general maintenance, including a change of membrane, and a pre-deployment calibration. This two-step calibration was used to correct the pCO₂ data as described by the supplier. Data collected after 2020-03-01 were not used because the Covid-19 pandemic prevented maintenance and the setup of a freshly calibrated sensor. As a result, algae became increasingly abundant, pulling pCO₂ down and further away from values calculated from $C_\mathrm{T}$ and $A_\mathrm{T}$. pCO₂was expressed at *in situ* temperature using the pCO2insi function of the R package seacarb v3.3.2 (Gattuso et al., 2023b).

## 2.4 pH

Two SeaFET Ocean pH sensors (Sea-Bird Scientific) were swapped on 2018-04-17, and 2019-09-02. While one was monitoring pH and temperature on the profiler, the other one was factory-calibrated. pH (volts) was measured continuously and data logged every minute. Calibration was performed as described by Bresnahan et al. (2014) using the functions sf_calib and sf_calc of the R package seacarb v3.3.2 (Gattuso et al., 2023b). Volts values measured below 8 m in each of the three deployment periods were converted to pH on the total scale (pHT). Field calibration samples for pH were collected using a Niskin bottle close to SeaFET within 15 min of measurement. pH was measured spectrophotometrically (Dickson et al., 2007) with purified m-cresol purple (purchased from Robert H. Byrne's laboratory, University of South Florida). A TRIS standard was measured 6 times. The deviation between the theoretical pH and pH measured ranged between -0.0033 and +0.0012 pH units (mean = -0.0015). The pHinsi function of the R package seacarb v3.3.2 (Gattuso et al., 2023b) was used to express pH at temperatures other than the measurement temperature from pH, salinity, and total alkalinity. The dissociation constants used are discussed below.

## 2.5 Data flow and quality insurance

Data collected at one minute frequency were assigned with quality flags following a series of quality tests (Table 2). Data with flags other than 1 (good data) were eliminated and outliers removed using despike function of the R package oce (Kelley and Richards, 2021) prior to calculating hourly averages.

## 2.6 Calculation of derived parameters of the carbonate system

The carb function of the R package seacarb v3.3.2 (Gattuso et al., 2023b) was used to calculate all parameters of the carbonate system from pairs of measured variables (e.g., $C_\mathrm{T}$ and $A_\mathrm{T}$, pCO₂ and $A_\mathrm{T}$, pH and $C_\mathrm{T}$), salinity, temperature and hydrostatic pressure. Total boron concentration was calculated from salinity (Lee et al., 2010). The following constants were used: $K_\mathrm{f}$ from Perez and Fraga (1987) and $K_s$ from Dickson (1990). The choice of the stoichiometric dissociation constants $K_1^*$ and $K_2^*$ is not obvious in polar oceans (Sulpis et al., 2020). Several sets of formulations were tested: Lueker et al. (2000), Millero et al. (2002), Papadimitriou et al. (2018) and Sulpis et al. (2020). Nutrient data (phosphate and silicate) were taken into consideration whenever available (van de Poll, unpublished data).

**Table 2.** Data quality flags.

| Flag | Description | Example |
|------|-------------|---------|
| 1 | Good data | Data not matching any of the other flags |
| 3 | Failing the date and time test | Data with impossible date (date outside of the project period) |
| 4 | Data not usable according to manufacturer | Data recorded during instrument flush or zeroing period |
| 7 | Failing the regional range test | Data out of range (e.g. salinity > 37) |
| 12 | Failing the spike test using the despike function of the R package oce Kelley and Richards (2021) with n=2 and k=5761 | Data assigned with NA as a result of the spike test |
| 15 | Instrument not deployed or operated | Data assigned with NA when the instrument is in maintenance |
| 16 | Data impacted by acid flush | Data during and after an acid flush (each day between 24:00 01:00) |
| 99 | Failing the final visual inspection | Data considered as outlier by visual inspection |

All parameters are reported at *in situ* temperature unless indicated otherwise. The average uncertainties of the derived carbonate parameters were calculated according to the Gaussian method (Dickson and Riley, 1978) implemented in the "errors" function of the R package seacarb v3.3.2 (Orr et al., 2018; Gattuso et al., 2023b). The uncertainties when using the $A_\mathrm{T}$-$C_\mathrm{T}$ pair are $\pm 2.7 \times 10^{-10}$ mol H$^+$ (about 0.015 units pH$_\mathrm{T}$), $\pm$ 15 $\mu$atm pCO$_2$ , and $\pm$ 0.1 unit for the aragonite and calcite saturation states. The maximum additional uncertainty associated with the unavailability of nutrient concentrations (P and Si) as input parameters is comparatively negligible (up to 0.0019 pH units, 1.5 $\mu$atm pCO$_2$ and 0.008 $\Omega_\mathrm{a}$ units).

## 2.7 Air-sea CO$_2$ flux

The instantaneous air-sea CO$_2$ fluxes were calculated as described by De Carlo et al. (2013) from measured pCO$_2$, atmospheric CO$_2$ measured at the Zeppelin station, also located at Ny-Ålesund (data downloaded on 2020-08-19 from https://gaw.kishou. go.jp/search/file/0054-6001-1001-01-01-9999, and the wind speed measured by the AWI at a height of 10 m (Maturilli, 2020). Two parameterisations between wind speed and the gas transfer velocity $k$(600) were used (Ho et al., 2006; Dobashi and Ho, 2023).

## 3 Dataset and discussion

The following sections describe the data set that is available at PANGAEA (https://doi.pangaea.de/10.1594/PANGAEA.957028) and provide a first analyses to demonstrate its usefulness.

### 3.1 Data availability

It is often mentioned that there are fewer observations in the Arctic Ocean than elsewhere but it is not the case for carbonate variables. We looked at $pCO_2$ records in the v2022 version of the SOCAT database (Bakker et al., 2016, 2022) and the dissolved inorganic carbon ($C_T$) records of the GLODAP v2.2022 database (Lauvset et al., 2022). About 12.4% of the SOCAT $pCO_2$ records and 11.1% of the GLODAP $C_T$ records are from the Arctic ocean as defined by the Organization (1953) which is only about 4.3% of the global ocean surface area. Coastal (bottom depth < 200 m) data are relatively well represented in both

products (24.3% and 11.1% of the SOCAT and GLODAP total Arctic data, respectively). The monthly distribution is however very uneven with 71.2% of the SOCAT $pCO_2$ data and 71% of the GLODAP $C_T$ data collected in four months of the year (June to September). Furthermore, few to very few data are available for December to March, including in coastal regions. To our knowledge, there is until today no high-frequency, multi-year time-series data.

The extreme environmental conditions prevailing at the study site incurred incidents such as interrupted supply of seawater

in the FerryBox due to frozen pipes or damages resulting from icebergs pounding on the field instruments. Resolutions of these incidents sometimes took weeks to months due to waiting for warmer temperatures to make deicing possible or to delays bringing technical staff, including divers, to repair damages. The study site was not accessible for extended periods of time during the Covid-19 pandemic, preventing discrete sampling resulting in data gaps. The lack of sensor maintenance sometimes generated data of poor quality which were eliminated, also generating gaps. Nevertheless, data were usable 50 to 76% of the

time during the period of measurement (Fig. 2A). Continuous $pCO_2$ and pH data are available throughout a composite year and well distributed across months, including in winter months (Fig. 2B). The total number discrete data available for $A_T$, $C_T$ and spectrophotometric pH is 195, 191 and 30. They are also well distributed across months (Fig. 2C).

### 3.2 Impact of the formulations of $K_1^*$ and $K_2^*$

Chen et al. (2015) found that the constants of Mehrbach et al. (1973) and Lueker et al. (2000) yield the best internal consistency

in Arctic waters over the temperature range of $-1.5 \leq T \leq 10.5$ °C and salinity range of $25.8 \leq S \leq 33.1$. They recommended the use of these constants. Sulpis et al. (2020) have shown that current estimates of $K_1^*$ and $K_2^*$ are inconsistent with measured $CO_2$ system parameters in cold oceanic region. The formulations of Lueker et al. (2000, L00), which are recommended by the community (Jiang et al., 2022), were derived in laboratory conditions with no temperature value below 2 °C. These formulations overestimates the stoichiometric dissociation constants at temperatures below about 8 °C (Sulpis et al., 2020). There are

several alternative formulations. Those of Millero et al. (2002, M02) and (Sulpis et al., 2020, S20) are based on large (> 900) field data that include cold temperature values. The formulations of Papadimitriou et al. (2018, P18), obtained in the laboratory, also cover cold temperatures.

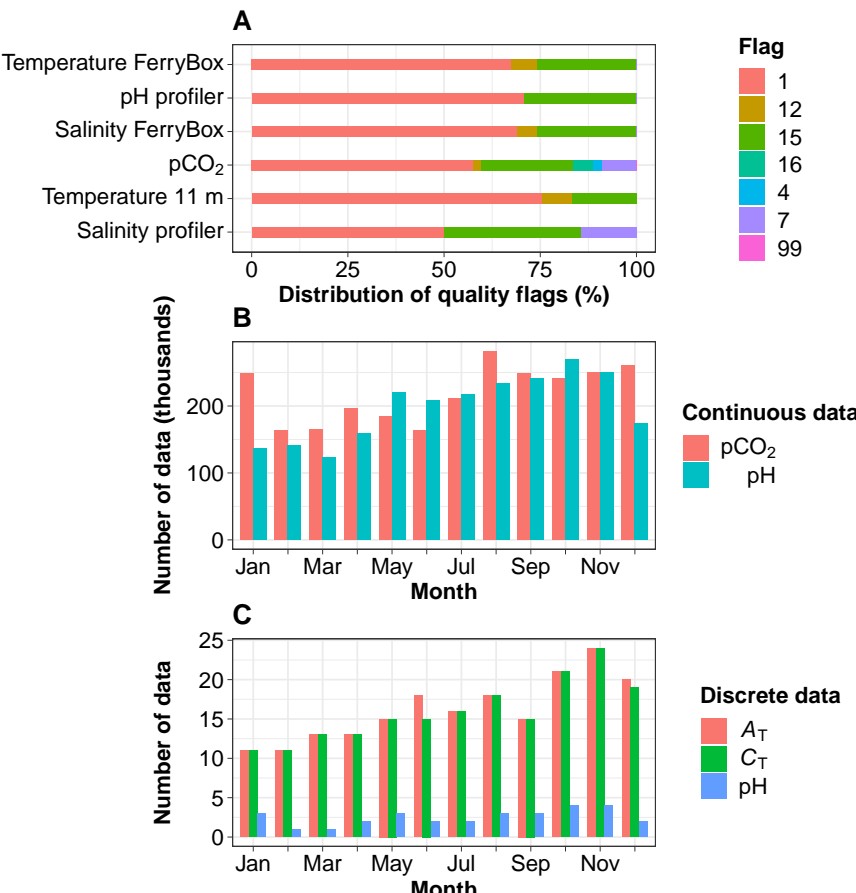

**Figure 2.** A: Distribution of the quality flags assigned to data collected every minute over the period July 2015 to December 2020, except for pH profiler sensor (SeaFET) which was set-up in August 2017. B: Monthly distribution of $pCO_2$ and pH data. C: Monthly distribution of discrete measurements of $A_T$, $C_T$ and spectrophotometric pH. Flags are defined in Table 2.

### 3.2.1 Using the pair $pCO_2$-$A_T$

For the pair $pCO_2$-$A_T$ (115 data pairs), it is the formulation of P18 which provides estimates of pH and $C_T$ closest to those obtained with L00 (Fig. 3). The absolute median difference between L00 and P18 are significantly smaller than the uncertainty estimated by error propagation for pH (0.001 vs 0.004 units) and $C_T$ (1.7 vs 3.6 $\mu$mol kg$^{-1}$). The formulation of M02 performs well for $C_T$ (1.5 vs 3.6 $\mu$mol kg$^{-1}$) but less well for pH (0.019 vs 0.004 units). The absolute median difference between L00 and S20 is similar to the uncertainty estimated by error propagation for $C_T$ (3.7 vs 3.6 $\mu$mol kg$^{-1}$) but is more than six times larger for pH (0.026 vs 0.004 units). For all formulations, the uncertainty for the saturation state for aragonite is negligible and smaller than that estimated with the propagation of errors.

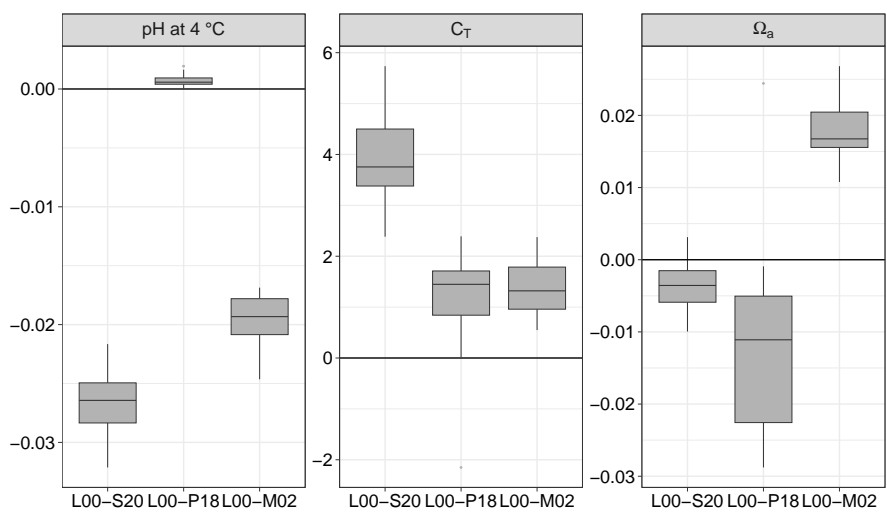

**Figure 3.** pH normalised at 4 °C, dissolved inorganic carbon ($C_T$) and saturation state of aragonite $\Omega_a$ calculated from pCO$_2$ and $A_T$ (115 data pairs): differences between the formulations for $K_1$ and $K_2$ of Lueker et al. (2000, L00) and those of Sulpis et al. (2020, S20), Papadimitriou et al. (2018, P18) and Millero et al. (2002, M02). Unit for $C_T$ is $\mu$mol kg$^{-1}$.

### 3.2.2 Using the pair $A_T$-$C_T$

The discrete values of $A_T$, $C_T$, salinity and temperature in the FerryBox were used to calculate pH using the same formulations for $K_1^*$ and $K_2^*$ as above (Fig. 3). Overall, the absolute median difference between the formulation of L00, on one hand, and S20, P18 and M02, on the other hand, is lowest with P18. The absolute median difference L00-P18 is small compared to the overall uncertainty estimated by error propagation: 0.004 vs 0.013 pH units and 3.1 vs 10.9 $\mu$atm pCO$_2$.

### 3.2.3 Measured pH vs pH calculated from pCO$_2$ and $A_T$

Here we compare pH measured spectrophotometrically with pH calculated from pCO$_2$ and $A_T$ using various formulations of $K_1^*$ and $K_2^*$ (Table 3). All pH values were normalized to a temperature of 4 °C. The absolute differences are up to 0.11 pH units. In general, all formulations overestimate spectrophotometric pH. pH calculated using the formulation of Lueker et al. (2000) is closer to measured pH, with a mean difference of -0.029 pH units. This difference is almost 9 times larger than the uncertainty for pH calculated from pCO$_2$ and $A_T$ estimated by error propagation (0.004 units). The next closer formulation is Papadimitriou et al.'s.

### 3.2.4 Measured pH vs pH calculated from $A_T$ and $C_T$

Here we compare pH measured spectrophotometrically with pH calculated from discrete measurements of $C_T$ and $A_T$ using various formulations of $K_1^*$ and $K_2^*$ (Table 4). All pH values were normalized to a temperature of 4 °C. The absolute differences

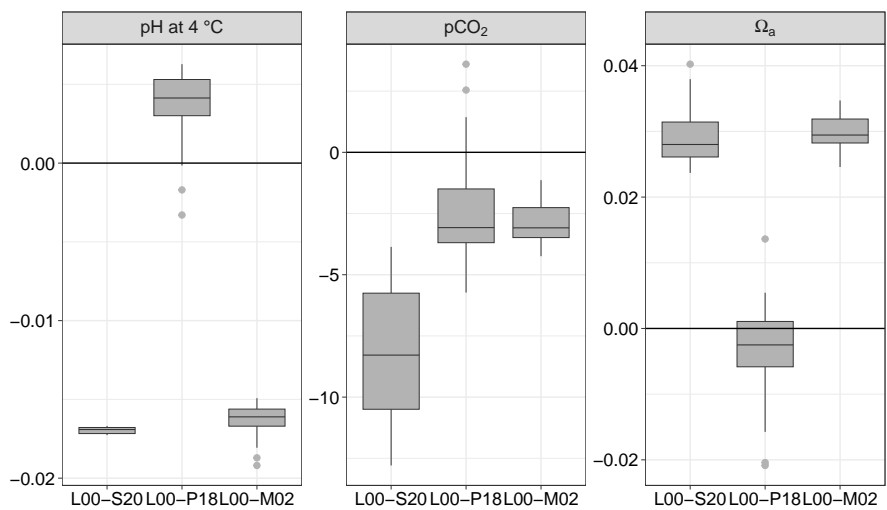

**Figure 4.** pH normalised at 4 °C, partial pressure of $CO_2$ and saturation state of aragonite $\Omega_a$ calculated from $A_T$ and $C_T$ (115 data pairs): differences between the formulations for $K_1$ and $K_2$ of Lueker et al. (2000, L00) and those of Sulpis et al. (2020, S20), Papadimitriou et al. (2018, P18) and Millero et al. (2002, M02). Units for $pCO_2$ is $\mu$atm.

**Table 3.** Difference between spectrophotometric pH and pH calculated with $pCO_2$ and $A_T$ using different formulations for $K_1^*$ and $K_2^*$. Q1 and Q3 are the first and third quartiles.

|         | Lueker et al. (2000) | Sulpis et al. (2020) | Papadimitriou et al. (2018) | Millero et al. (2002) |
|---------|---------------------|---------------------|----------------------------|----------------------|
| Minimum | -0.086              | -0.108              | -0.083                     | -0.110               |
| Q1      | -0.036              | -0.069              | -0.042                     | -0.056               |
| Median  | -0.026              | -0.060              | -0.033                     | -0.046               |
| Mean    | -0.029              | -0.059              | -0.032                     | -0.049               |
| Q3      | -0.020              | -0.045              | -0.019                     | -0.041               |
| Maximum | 0.012               | -0.029              | 0.000                      | -0.007               |

can be as high as 0.133 pH units. In general, all formulations overestimate spectrophotometric pH. pH calculated using the formulations of Lueker et al. (2000) and Papadimitriou et al. (2018) are closer to measured pH, with absolute median differences of -0.007 pH units. This difference is much smaller than the uncertainty for pH calculated from $A_T$ and $C_T$ according to seacarb (0.017). The mean differences found with the other formulations are slightly lower than the uncertainty for pH calculated from
185 $A_T$ and $C_T$ according to seacarb.

**Table 4.** Difference between spectrophotometric pH and pH calculated with $A_\mathrm{T}$ and $C_\mathrm{T}$ using different formulations for $K_1^*$ and $K_2^*$. Q1 and Q3 are the first and third quartiles.

|  | Lueker et al. (2000) | Sulpis et al. (2020) | Papadimitriou et al. (2018) | Millero et al. (2002) |
|---|---|---|---|---|
| Minimum | -0.112 | -0.133 | -0.113 | -0.129 |
| Q1 | -0.032 | -0.048 | -0.030 | -0.049 |
| Median | -0.007 | -0.027 | -0.007 | -0.024 |
| Mean | -0.015 | -0.034 | -0.014 | -0.031 |
| Q3 | 0.007 | -0.015 | 0.007 | -0.010 |
| Maximum | 0.081 | 0.064 | 0.087 | 0.065 |

In conclusion, the formulations of Lueker et al. (2000) and Papadimitriou et al. (2018) have similar performances with our dataset and generally perform better than those of Millero et al. (2002) and Sulpis et al. (2020). The formulation of Papadimitriou et al. (2018) is seldom used and the *de facto* standard has become the formulations of Lueker et al. (2000), which we have used in the present study.

### 3.3 Impact of nutrient concentrations

Phosphate ($PO_4$) and silicate (Si) contribute to total alkalinity. Changes in their concentration can significantly affect calculations of the carbonate chemistry. The impact on our calculations was checked with a time series of nutrients comprising 90 phosphate and 133 silicate data kindly provided by van de Poll (unpubl. data). At the study site, the concentrations of $PO_4$ and Si vary by a factor 10 along a composite year. They range between 0.07 and 0.69 $\mu$mol kg$^{-1}$ for $PO_4$ and between 0.42 and 4.7 $\mu$mol kg$^{-1}$ for Si. In our dataset, disregarding the nutrient concentrations does not generate large differences in the derived parameters. Using the pCO$_2$-$A_\mathrm{T}$ pair of variables, the absolute differences in pH, $C_\mathrm{T}$ and $\Omega_\mathrm{a}$ are respectively 0.0001 unit, 0.7 $\mu$mol kg$^{-1}$ and 0.001. With the $C_\mathrm{T}$-$A_\mathrm{T}$ pair, the absolute differences in pH, pCO$_2$ and $\Omega_\mathrm{a}$ are 0.002 units, < 1.5 $\mu$atm and <0.01.

### 3.4 Relationship between total alkalinity and salinity

The relationship between the total alkalinity ($A_\mathrm{T}$) and salinity (S) is good (Fig. 5A). The equation of the ordinary least square linear regression is $A_\mathrm{T} = 47.6 + 643 \times S$ ($r^2 = 0.81, N = 181$). The root mean square error (rmse) is 16.8 $\mu$mol kg$^{-1}$. Hunt et al. (2021) reported significant seasonal shifts in linear $A_\mathrm{T}$ vs S relationships on the East coast of the USA, demonstrating potential problems with any single linear model for the retrieval of $A_\mathrm{T}$ from salinity. There is no obvious seasonal shift in our data set. Splitting the data and regressing separately with salinity values below and above 34.5, as done by Nondal et al. (2009) for Nordic open ocean waters, does not prove useful (data not shown). It degrades $r^2$ (0.74 and 0.3 vs 0.81), and degrades or

marginally improves the rmse (19 and 13.6 vs 16.8 $\mu$mol kg$^{-1}$). The relationship above was therefore used to estimate $A_T$ from salinity.

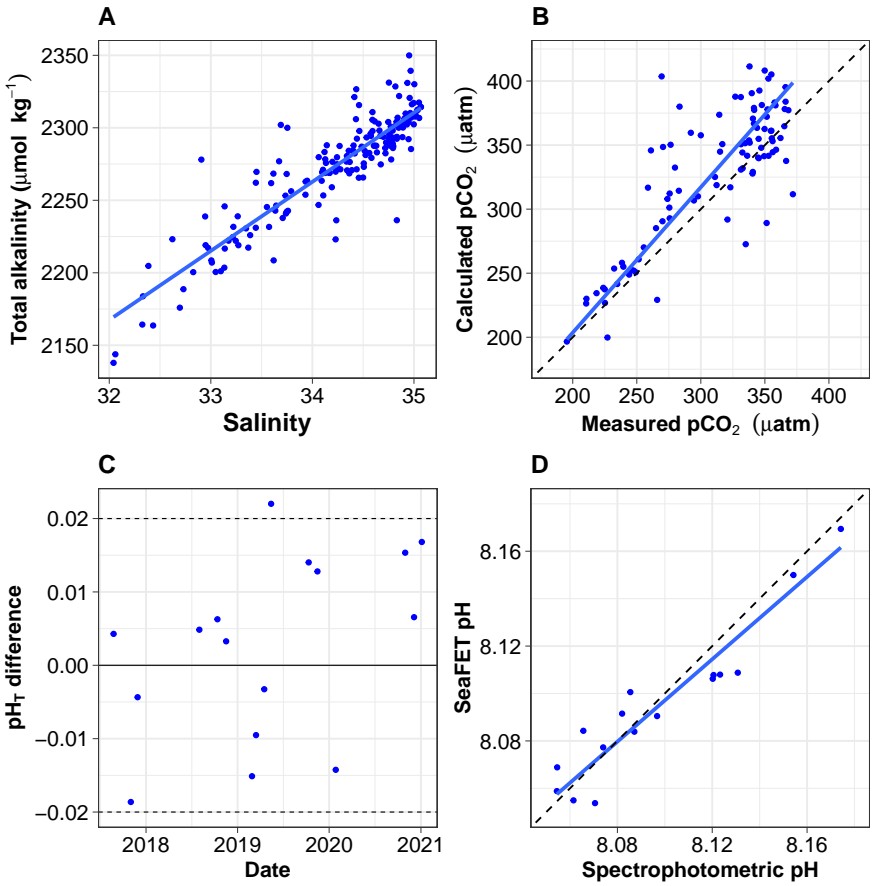

**Figure 5.** A: Relationship between discrete total alkalinity and salinity; the regression line is estimated using ordinary least square regression. B: pCO$_2$ calculated from $A_T$ and $C_T$ vs pCO$_2$ measured using the Contros sensor. All data are normalized at *in situ* temperature. The black dotted line is the 1:1 line while the blue solid line is calculated using a major axis regression. C: Offset (total scale) between spectrophotometric measurements of pH and the calibrated SeaFET pH time series. D: SeaFET pH vs spectrophotometric pH. All data on the total scale and normalized at *in situ* temperature. The black dotted line is the 1:1 line while the blue solid line is calculated using a major axis regression.

### 3.5 Consistency of measured vs calculated pCO$_2$

The relationship between the measured and calculated pCO$_2$ (blue line) is relatively poor (Fig. 5B). The slope is 1.12 and its 95% confidence interval includes 1. The equation of the major axis regression is: Calculated pCO$_2$ ($\mu$atm) $= -23.5 + 1.14 \times$ Measured pCO$_2$ ($r^2 = 0.66, N = 95$).

### 3.6 Calibration of SeaFET pH sensors and consistency of measured vs calculated pH

The offset between the spectrophotometric reference samples and the calibrated SeaFET pH time series must be between -0.2 and 0.2 pH units (McLaughlin et al., 2017). The mean offset was $\pm$ 0.0026 units, with only one data point outside the recommended range, indicating a high-quality pH dataset (Fig. 5C).

The relationship between spectrophotometric pH and SeaFET pH (blue line) is relatively good (Fig. 5D). The slope is 0.869 and its 95% confidence interval includes 1. The equation of the major axis regression is: SeaFET pH $= 1.06 + 0.869 \times$ spectrophotometric pH ($r^2 = 0.89, N = 16$).

### 3.7 Time series and monthly distribution of key parameters

The changes in salinity, temperature, partial pressure of $CO_2$, pH and total alkalinity are shown in Fig. 6A-E and monthly box plots in Fig. 6F-J. Salinity below 8 m is highest in the spring and lowest in the fall with monthly median values of 35 and 33.3, respectively. Positive salinity extremes (values > 90th percentile) mostly occur in March-June, presumably due to intrusion of seawater from the open sea. Negative salinity extremes (values < 10th percentile) are mostly observed in the summer (defined here as 3 months from June to August) and early fall, periods during which melting sea ice, calving glaciers and numerous streams release freshwater in the coastal zone. Temperature at 11 m is lowest in February and highest in August with monthly median values of -0.1 and 6.1 °C. Total alkalinity exhibits relatively large changes with lower values in the summer and early fall. Similar and even larger declines have been reported in Spitsbergen fjords (e.g., Koziorowska-Makuch et al., 2023). They are the result of freshwater input which generally has a diluting effect and lowers $A_\mathrm{T}$ in surface waters .

$pCO_2$ at 11 m is almost always lower than 400 $\mu$atm with low values during and following the spring phytoplankton bloom and high values in winter. The relative importance of thermal and non-thermal (physical and biological) processes in controlling $pCO_2$ was investigated as described by Takahashi et al. (2002). The thermal/non-thermal ratio is lower than 1 for 9 months of a composite year, indicating that non-thermal drivers exert a greater control than temperature (Fig. 7). The ratio is above 1, hence thermal control is predominant, only in the three winter months of December, January and February.

### 3.8 Depth distribution

There is no depth profile of the variables in the usual sense as the REMOS profiler made stops for 24 h at specific depths to assess the biota in the water column (Fischer et al., 2017). However, the depth distribution of the median monthly salinity, temperature and density provide useful information (Fig. 8). Salinity in the bottom layer (8 to 12 m) is up to 0.9 units higher than in the surface layer (0 to 4 m) in summer, 0.6 units lower in December and relatively similar in both layers at other times. Temperature is lower by up to 2 °C in the deep layer than in the surface layer from January to October and higher by up to 0.3 °C in November and December. Seawater density is always higher in the bottom than in the surface layer (up to 1.2 kg m$^{-3}$ in July). The 12 m high water column is therefore generally stratified. This is a well-known feature, particularly in the Arctic due to low-salinity surface waters (Dong et al., 2021; Miller et al., 2019).

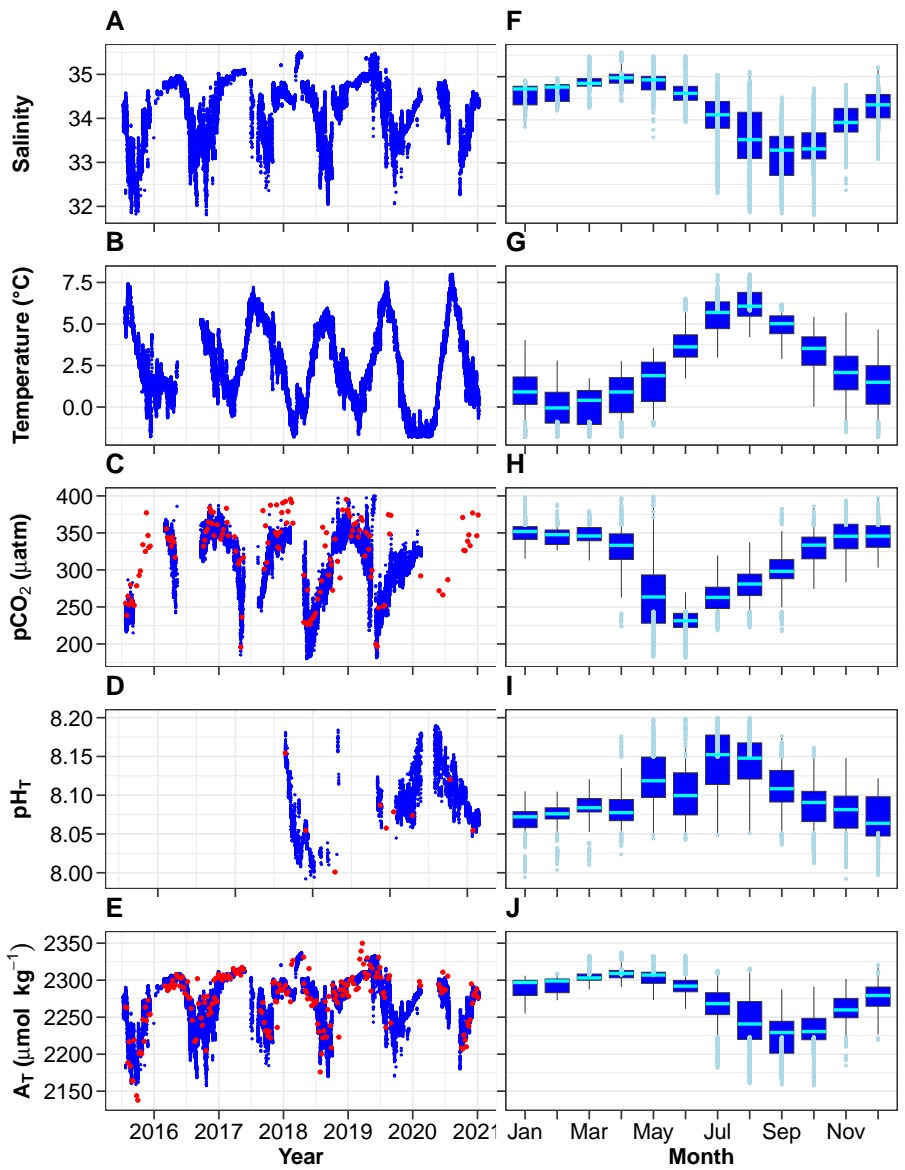

**Figure 6.** A-E: Time-series (A-E) and monthly distribution (F-J) of key environmental parameters (hourly means). Panel C: pCO₂ measured (red) and calculated using $A_T$ and $C_T$ (blue). Panel D: $pH_T$ measured (red) and calculated using $A_T$ and $C_T$ (blue). Panel E: $A_T$ measured by potentiometric titration (red) and calculated from the $A_T$—salinity relationship (blue). In panels F-J, the cyan lines indicate the medians, boxes show the first and third quartiles and the interquartile range, whiskers extend to the 5–95th percentiles. The light blue circles highlight values above the 90th percentile and below the 10th percentile.

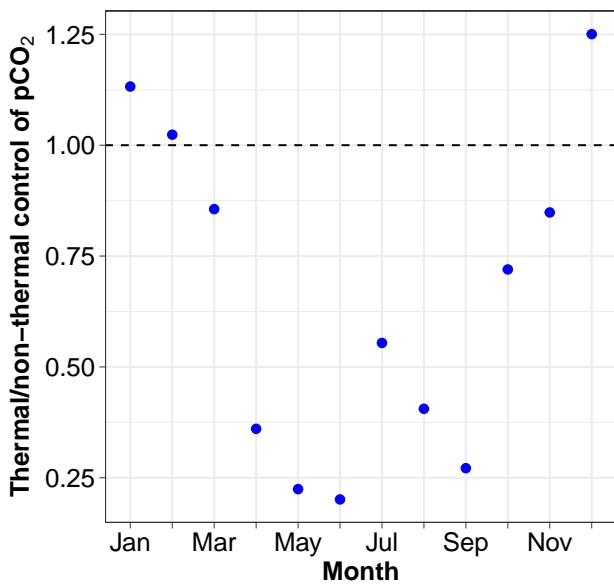

**Figure 7.** Ratio of the thermal vs non-thermal control of pCO$_2$.

### 3.9 Air-sea CO$_2$ fluxes

pCO$_2$ of seawater pumped at 11 m depth was measured in the FerryBox. This is not the best arrangement to estimate air-
sea CO$_2$ fluxes considering the fact that the water column was sometimes stratified as shown by vertical gradient of salinity,
temperature and density (Fig. 8). This is known to have consequences on the air-sea CO$_2$ flux. pCO$_2$ is generally higher in the
bottom layer than in the surface layer (note that no data is available in May, June and July).

To estimate air-sea CO$_2$ fluxes, pCO$_2$ can also be calculated using water-column variables measured or estimated from
sensors attached to the REMOS device: SeaFET pH, temperature, salinity and salinity-derived total alkalinity. At *in situ* tem-
perature, the vertical gradient is within $\pm 4$ $\mu$atm, except in April where it is more than 40 $\mu$atm (Fig. 9A). Normalising pCO$_2$
at 4 °C (Fig. 9B) reduces the April difference from -45 to -22.6 $\mu$atm, indicating that the vertical gradient is partly driven by
temperature.

For the 9 months when data are available, monthly median pCO$_2$ normalized at *in situ* temperature at 11 m vs 0-4 m are well
correlated ($r^2 = 0.81$) but pCO$_2$ is higher at the surface than at 11 m, with a median difference of 17 $\mu$atm (Fig. 10).

The air-sea CO$_2$ flux estimated from pCO$_2$ at 11 m is negative, indicating a CO$_2$ influx from the atmosphere, every month
of a composite year (Fig. 11). The gas exchange coefficient $k$ is notoriously difficult to measure. It is often parameterised by
wind speed which is known to work well in deep waters offshore (Ho et al., 2006). In shallow areas, parameters other than
wind speed become important. Dobashi and Ho (2023) proposed a formulation which might work better in wind-fetch-limited
environments. Here we are bracketing the air-sea CO$_2$ flux using these two parameterisations. The annual air-sea flux ranges

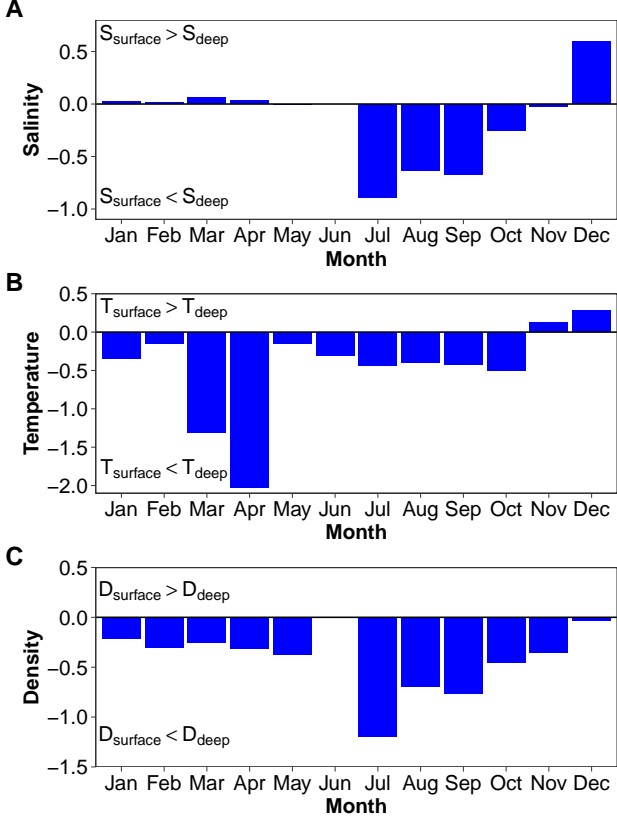

**Figure 8.** Vertical gradients calculated using the median monthly values of salinity (a), temperature (b) and density (c). "Surface" is 0 to 4 m and "deep" is 8 to 12 m.

from -10.2 to -20.2 mol $CO_2$ $m^{-2}$ $yr^{-1}$, respectively with the formulations of Dobashi and Ho (2023) and Ho et al. (2006). Correcting for the fact, discussed above, that surface $pCO_2$ is higher than $pCO_2$ at 11 m above leads to fluxes of -16.8 and -9 mol $CO_2$ $m^{-2}$ $yr^{-1}$ with the two parameterisations.

These values are in good agreement with the literature. The Arctic Ocean stands out as the region with the strongest $CO_2$ uptake per unit area during the period 1985–2019, with $-8.6 \pm 0.4$ mol $m^{-2}$ $yr^{-1}$ for the open sea and $-5.6 \pm 0.4$ mol $m^{-2}$ $yr^{-1}$) for the continental shelf margins (Chau et al., 2022). Air-sea $CO_2$ flux range from -4 to -86 mol $m^{-2}$ $d^{-1}$ (Bates and Mathis, 2009; Bates et al., 2011; Rysgaard et al., 2012). For example, the surface waters of the entire Godthåbsfjord (west Greenland) and adjacent continental shelf are undersaturated in $CO_2$ throughout the year (Meire et al., 2015). The average annual $CO_2$ uptake within the fjord is estimated to be 5.42 mol $m^{-2}$ $yr^{-1}$ , indicating that the fjord system is a strong sink for $CO_2$.

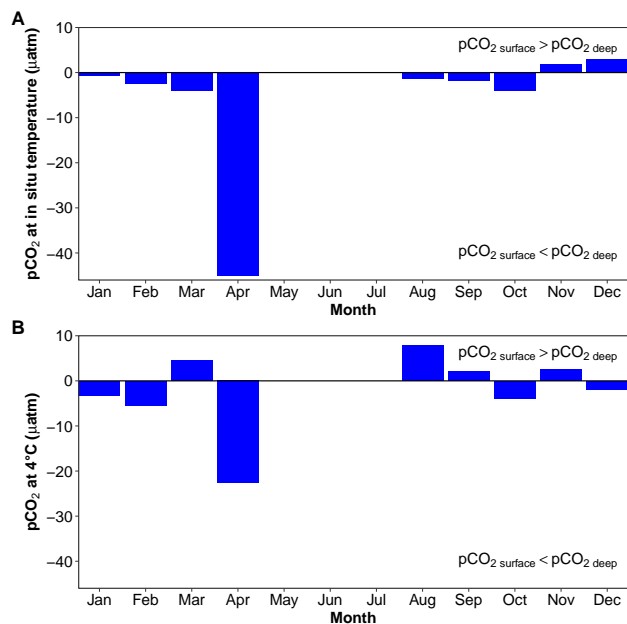

**Figure 9.** Vertical gradients estimated using the median monthly values of $pCO_2$ at *in situ* hydrostatic pressure, calculated from $A_T$ (using the $A_T$ vs S relationship) and SeaFET $pH_T$ using the R package seacarb (Gattuso et al., 2023b). A: $CO_2$ at *in situ* temperature; B: $pCO_2$ normalised at 4 °C. "Surface" is 0 to 4 m while "deep" is 8 to 12 m. Data are missing in May to July because no surface pH data is available during this period.

### 3.10   Saturation state of $CaCO_3$

The saturation state of $CaCO_3$ is subject to large interannual changes (Fig. 12). $\Omega_a$ never becomes lower than 1. It ranges between 1.4 in winter to 3 in summer.

## 4   Conclusion

Although measurements of the carbonate system have increased significantly in the Arctic Ocean, there is still a lack of high-frequency time series, also in the coastal zone. Autonomous time-series measurements in the Arctic involve a number of

challenges related to remoteness and harsh environment (Fischer et al., 2020). The most serious incidents our study faced related to system damages from iceberg collisions as well as frozen tubes delivering sea water to the land based measuring system. The remoteness and harsh environmental conditions made maintenance difficult especially during the polar winter and led to a discontinuous dataset. Even though we planned this dataset to become a real long-term dataset, unfortunate non-technical circumstances brought this time series to an end, preventing the assessment of interannual variability. Nevertheless,

it is unique by its high (hourly) frequency, coverage of all seasons, and duration (over 4 years).

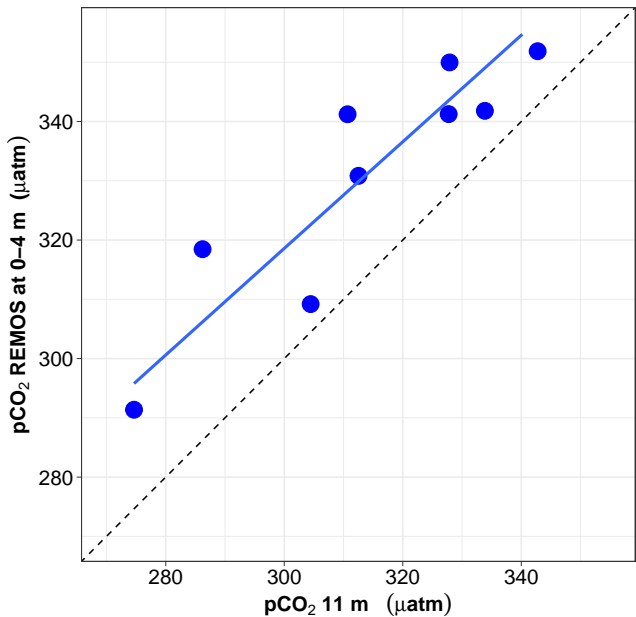

**Figure 10.** Relationship between surface pCO$_2$ [0-4 m] (estimated from pH and salinity-derived $A_T$) and pCO$_2$ at 11 m. Both values are expressed at *in situ* temperature. The mean and median difference between the two pCO$_2$ are about 17 $\mu$atm. The major axis regression line is shown in blue whereas the 1:1 relationship is depicted by a black dotted line.

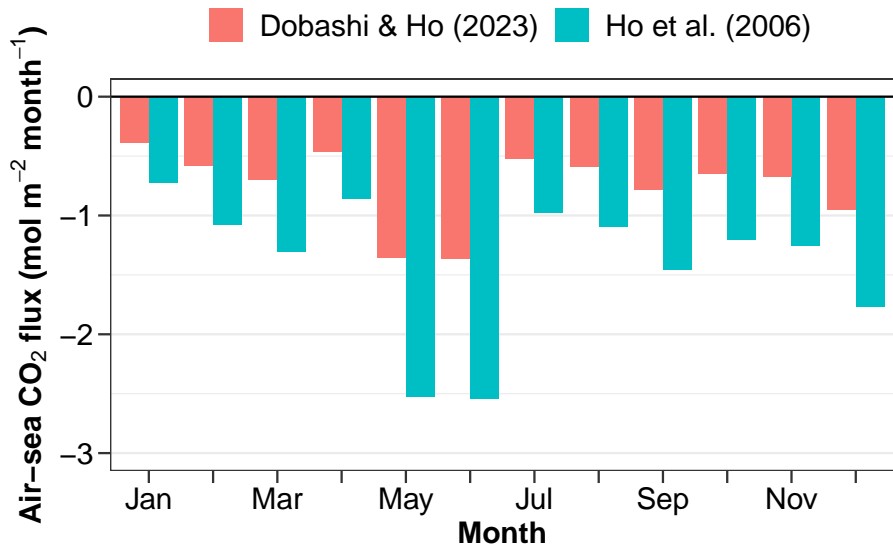

**Figure 11.** Air-sea CO$_2$ flux estimated using corrected pCO$_2$ values and the parameterisations of the gas exchange coefficient by wind speed of Ho et al. (2006) and Dobashi and Ho (2023).

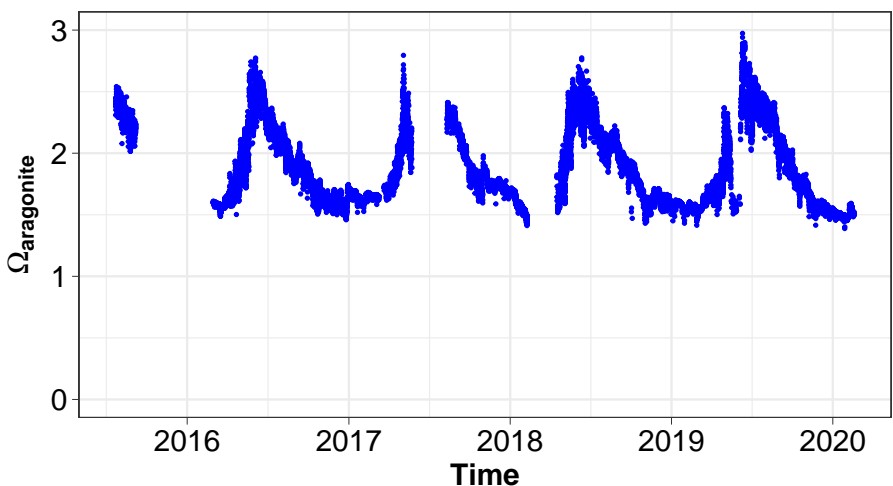

**Figure 12.** Time series of the aragonite saturation state calculated using pCO$_2$ and salinity-derived total alkalinity as input parameters.

The final data product provides information on a series of key questions on the dynamics and carbon cycling in a high-Arctic fjord. Several have been discussed above. Our results show that (1) the choice of formulations for calculating the dissociation constants of the carbonic acid remains unsettled, (2) the 12-m high water column is consistently stratified most of the time, (3) the calcium carbonate saturation state is subject to large seasonal changes but never reaches undersaturation, (4) this coastal site is a large CO$_2$ sink.

### 4.1 Data availability

Data are available on Zenodo during the review process (Gattuso et al., 2023b): https://doi.org/10.5281/zenodo.7714954. The final version will be published in the World Data Center PANGAEA after acceptance of the paper (Gattuso et al., 2023a): https://doi.pangaea.de/10.1594/PANGAEA.957028.

The csv file "AWIPEV-CO2_v1.csv" comprises the following variables:

- Continuous variables (hourly means):

    - Date/time [UTC+0]: date and time at UTC+0
    - pressure_profiler [dbar]: hydrostatic pressure (profiler)
    - salinity_PSS78_profiler [unit]: salinity in situ (profiler)
    - salinity_PSS78_ferrybox [unit]: salinity (FerryBox)
    - temperature_ITS90_11m [°C]: temperature *in situ* (static at 11 m)

- temperature_ITS90_profiler [°C]: temperature *in situ* (profiler)

- temperature_ITS90_ferrybox [°C]: temperature (FerryBox)

- temperature_ITS90_seafet_profiler [°C]: temperature SeaFET (profiler)

- pco2_insitu_temperature_ferrybox [uatm]: partial pressure of $CO_2$ (FerryBox)

- ph_insitu_temperature_profiler [total scale]: pH *in situ* at *in situ* temperature (profiler)

– Discrete variables:

- ta_discrete [$\mu$mol kg$^{-1}$]: total alkalinity *in situ* (discrete)

- dic_discrete [$\mu$mol kg$^{-1}$]: dissolved inorganic carbon *in situ* (discrete)

- ph_discrete [total scale]: spectrophotometric pH *in situ* at *in situ* temperature (discrete)

*Author contributions.* JPG conceived the project. PF led the sensor implementation and underwater sensor maintenance. SA and PF maintained the FerryBox system, the instrumentation and the continuous data transfer. SA and JPG led data processing and analysis. JPG led the analysis and writing with contributions from SA and PF. JPG wrote the draft and coauthors contributed text and edits.

*Competing interests.* The authors declare that they have no conflict of interest.

*Acknowledgements.* Thanks are due to Robert Schlegel (Sorbonne Université) for help with data analyses and upload, Li-Qing Jiang (NOAA) for advice about variable names, David Ho for input on gas exchange coefficients, and Mohammed Khamla for assistance with graphics. We also thank Leif Anderson, Yuanxu Dong and Nicolas Metzl for their constructive reviews which significantly improved the paper. We are extremely grateful to the AWIPEV staff who made the continuous operation of the underwater observatory in such a remote location possible almost flawlessly until 2020. Such gratitude cannot be extended to the staff on duty in 2020. We thank the numerous divers from
the AWI Centre for Scientific Diving for their invaluable assistance during the maintenance trips. The $A_T$ and $C_T$ data used in this study were analyzed at the SNAPO-$CO_2$ service facility at LOCEAN laboratory and supported by CNRS-INSU and OSU Ecce-Terra. We are indebted to Willem H. van de Poll who kindly provided nutrient data. We thank Ove Hermansen, Cathrine Lund Myhre, and Stephen Platt at Norwegian Institute for Air Research (NILU) for their assistance with atmospheric $CO_2$ data from the Zeppelin observatory, as well as the Integrated Carbon Observing System (ICOS)-Norway, Norwegian Research Council project NFR-207587, and the Norwegian Environment
Agency. Atmospheric $CO_2$ data from Zeppelin is available from EBAS: http://ebas.nilu.no. This work has been supported by the Coastal Observing System for Northern and Arctic Seas (COSYNA), the two Helmholtz large-scale infrastructure projects ACROSS and MOSES, the French Polar Institute (IPEV) as well as the European Union's Horizon 2020 research and innovation projects Jericho-Next (No 871153 and 951799), INTAROS (No 727890) and FACE-IT (No 869154).

**Appendix A: Related datasets**

Longer (2012-2021) datasets are available for salinity and temperature (Fischer and colleagues). They are stored in the open access repository PANGAEA:

- 2012: https://doi.org/10.1594/PANGAEA.896828

- 2013: https://doi.org/10.1594/PANGAEA.896822

- 2014: https://doi.org/10.1594/PANGAEA.896821

- 2015: https://doi.org/10.1594/PANGAEA.896771

- 2016: https://doi.org/10.1594/PANGAEA.896770

- 2017: https://doi.org/10.1594/PANGAEA.896170

- 2018: https://doi.org/10.1594/PANGAEA.897349

- 2019: https://doi.org/10.1594/PANGAEA.927607

- 2020: https://doi.org/10.1594/PANGAEA.929583

- 2021: https://doi.org/10.1594/PANGAEA.950174

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
