# Peer review of "High-frequency, year-round time series of the carbonate chemistry in a high-Arctic fjord (Svalbard)"

_Earth System Science Data, 2023_

## Referee Comment (RC2)

This work provides valuable long-term carbonate observations and guidance on instrument setup, data processing and quality control for the coastal ocean measurements of carbonate species. The dataset would be a great contribution to the Arctic carbon cycle community and the method will be well-welcomed by the coastal ocean community. I think it is suitable to be published in ESSD upon resolving the minor corrections below.

By the way, my expertise mainly allows me to comment on the pco2 and CO2 flux-related contents. Please refer to the other reviewer's comments for the remaining parts. Also, I am not a native English speaker, but the writing looks good to me.

General and specific comments:

Line 6: Is it possible to use the data present in this study to back-calculate the dissociation constants?

Line 7: '…remains unsettled for Arctic waters'. How representative the water at the measured location for the entire Artic waters?

Line 7: Does the stratification related to the ocean depth? If no, suggest removing 'despite the shallow depth'. Also, I did not see any discussion about this in the main text.

Line 10: in the main text, the value is 17. Keep consistency.

Line 12: 'are understood the least'. Why the least? Any reference for this?

I was thinking the Antarctic and Southern Ocean are less understood because of the remote and limited measurements.

Line 20: I am curious why the Arctic SST increasing rate right now is not significantly higher than other regions considering the greatest future warming?

Line 48-49: Fig. 1 A, B, C. The figure caption uses the capital A, B, C to represent the subplots. Keep consistency.

Line 50-51: I am wondering if this sulfuric acid will influence the pH and carbonate measurements. May quickly dilute by the water mixing?

Figure 1: worth to check all the figure captions. Here a, b, c, d should be 1, 2, 3, 4 I think. In addition, add (C): Svalbard (A), Kongsfjorden and Ny-Ålesund (B), and (C) observational set-up…

Line 63: 'The number of outliers discarded was 38 and 41'. How many observations are in total?

Line 98: consider removing '…' in the bracket.

Line 114: 'The gas exchange parameterization as a function of wind speed of Ho et al. (2006) was used'. I like Ho et al. (2006), but not sure if Ho et al. (2006) parametrization is the best here at a coastal environment. First, the (Ho et al., 2006) was derived from the open ocean (Southern Ocean) environment, while the (Nightingale et al., 2000) was derived from the coastal sea (the North Sea), which may be better here. Second, the K in the very coastal area (very shallow seawater) might be different from the K in a relatively open ocean (see (Yang et al., 2019)). May not need to change the $K$ parameterisation here, but worth to add this information.

Line 125: not clear to me which time-series data is mentioned here.

Figure 2: B: Monthly; C: Monthly

Figure 4: Here I did not see pco2 plot. Does the middle panel represent pco2? In addition, 'aragonite Ωa calculated from AT and AT', two AT, should be a typo.

Line 173-175: Looks to me that the performance of Papadimitriou et al. (2018) and Lueker et al. (2000) is similar. Worth to explain why 'the formulations of Papadimitriou et al. (2018) performs better on our data set'.

Line 175: remove one right bracket.

Line 195: 'The relationship between the measured and calculated pCO2 (blue line) is relatively poor'. What is the reason of this?

Figure 5: add A after 5.

Figure 6: explain the blue and red dots in subplots A-E. In addition, please check here 'In panels F-J, the orange lines [to add] indicate the medians, boxes show the first and third quartiles and the interquartile range'. Some typos.

Line 222-223: 'Temperature is lower by up to 2 ∘C in the deep layer from January to October and higher by up to 0.3 ∘C in November and December.' Lower than the surface layer, higher than the surface layer?

Figure 8: Any thoughts on why the salinity and temperature at surface is higher than at deeper water in December? Caption: A, B, C, not a, b, c.

Line 232: Figure 5B shows that the calculated pco2 disagree with the directly measured pco2. Not sure how good the calculation here is. Consider providing the uncertainty of the calculated pco2.

Line 236 & Figure 10: here in the text, surface pco2 < deep pco2, but figure 10 shows opposite. Please check and revise.

Line 239: It would be interesting to include some discussion about the surface pco2 > deep pco2

scenario in December.

Line 240-242: Here is quite confusing. Does the value 20 missed a '-' (i.e., -20?)? 'Correcting for the underestimation of 17 μatm…', I think it is overestimation? But if the uncorrected flux is -20 mol m−2 yr−1, after the 17 uatm correction, the flux should be higher than 20 in magnitude. Please double check and clarify carefully, because this is quite important for the conclusion.

Line 243-245: I guess here try to mention the entire Arctic Ocean carbon uptake. But I am wondering how representative for the Arctic Ocean this location is? I feel like this is just a coastal ocean, which is likely different from the relatively open seawater and also different from other coastal oceans.

-Yuanxu Dong

---

## Author Comment (AC1)

**Reply to referee #1 of the manuscript "High-frequency, year-round time series of the carbonate chemistry in a high-Arctic fjord (Svalbard)"**

Jean-Pierre Gattuso et al.

June 5, 2023

We are very grateful to referee Leif Anderson for his constructive comments which greatly improved the manuscript. Below is a point-by-point reply (**RC**: referee comment; **AR**: author reply)

**RC:** This manuscript presents a very valuable high-frequency data set of carbon system relevant parameters covering several years in the surface water of an Arctic fjord system. Never before has it been possible to observe the evolution of climate relevant parameters as pCO2 and saturation state of aragonite in all seasons with this high time resolution. These data will set a very useful reference point for other studies of the carbon system in the Arctic Ocean.
Hence, well deserve to be published, but can be improved by making some minor changes as specified below.

 **AR:** Thank you.

**RC:** Section 2.4. The SeaFET sensor is pressure sensitive and it therefore valuable to give information on how long the profiling system was kept at the depth before recording. On line 2018 it is given as 24 hours but this information would be valuable also here.

 **AR:** The sensing element of the seaFET sensor is solid state and therefore insensitive to pressure (according to the supplier and within specifications). In any case, the median time spent by the sensor at each depth interval was 6 h rather than 24 h. the manuscript has been revised accordingly.

**RC:** Line 106. The uncertainties given, I guess, is a result of analytical imprecision of the input parameters, but no information of these imprecisions are given in 2.2. Please do that, and also give the accuracy, which is as important for the computation of the other C-parameters. The same should be done for the discrete pH measurements.

 **AR:** The uncertainties given are the results of the propagation of the analytical uncertainties of $A_T$ and $C_T$ as well as the uncertainties of salinity, temperature, total boron, and of the 7 key dissociation constants. We agree that the analytical accuracy and precision of $A_T$ and $C_T$ should be given in Section 2.2 and have done so in the revised version of the manuscript:

 *The average accuracy of $C_T$ and $A_T$ measurements was 2.6 and 3 µmol $kg^{-1}$ , respectively, compared to seawater certified reference material (CRM) provided by A. Dickson (Scripps Institution of Oceanography). Repeatability of replicate samples was better than 3 µmol $kg^{-1}$.*

 Concerning the pH measurement of discrete samples, a TRIS standard was measured 6 times. The deviation between the theoretical and measured pH ranged between -0.0033 and +0.0012 pH units (mean = -0.0015). This information is provided in the revised version of the manuscript.

**RC:** Section on lines 173-175. I cannot follow this text. From the figures 3 & 4 as well as tables 3&4 I only see marginal differences in the results when using Lueker et al (2000) and Papadimitriou et al. (2018). The authors need to better describe what they mean.

    **AR:** The manuscript was revised accordingly as follows.

    *In conclusion, the formulations of Lueker et al. (2000) and Papadimitriou et al. (2018) have similar performances with our dataset and generally perform better than those of Millero et al. (2002) and Sulpis et al. (2020). The formulation of Papadimitriou et al. (2018) is seldom used and the de facto standard has become the formulations of Lueker et al. (2000), which we have used in the present study.*

**RC:** Fig. 6. Add the orange lines as noted in the legend. Also specify what the red dots in C, D and E are. I guess that for D and E it is the measured values in water samples and thus the blue dots in E must be computed; from salinity? For pCO2 the situation must be different as it was only measured by the ferry box. Please give information on this.

    **AR:** Lines were added. We agree that the legend was incomplete. It now reads :*A-E: Time-series (A-E) and monthly distribution (F-J) of key environmental parameters (hourly means). Panel C: $pCO_2$ measured (red) and calculated using $A_T$ and $C_T$ (blue). Panel D: $pH_T$ measured (red) and calculated using $A_T$ and $C_T$ (blue). Panel E: $A_T$ measured by potentiometric titration (red) and calculated from the $A_T$—salinity relationship (blue). In panels F-J, the cyan lines indicate the medians, boxes show the first and third quartiles and the interquartile range, whiskers extend to the 5–95th percentiles. The light blue circles highlight values above the 90th percentile and below the 10th percentile.*

**RC:** In line 207 it reads "salinity below 8 m .." while it in line 211 reads "Temperature at 11 m ..". I hope that all the high time resolution data, except that of pH, is from the ferry box. If not any comparison is prone to uncertainties in water masses variability. Please specify the depth of sampling in detail.

    **AR:** The salinity shown in Fig. 6 is indeed salinity below 8 m (the label of the Y-axis has been revised accordingly). The reason is that the salinity sensor in the FerryBox had some failures. The gaps were filled by salinity values measured with the *in situ* CTD when the REMOS was below 8 m. Such gap filling was not performed for temperature which warms by about 1°C before reaching the FerryBox.

    The following paragraph was added in the Material and Methods section: *The salinity (conductivity) sensor in the FerryBox had some failures. The gaps were filled by salinity values measured with the in situ CTD when the REMOS was below 8 m. Such gap filling was not performed for temperature which warms by about 1°C before reaching the FerryBox.*

**RC:** The first paragraph of 3.9 needs to be looked over. The data that is not available in May to July are pH, not the ferry box pCO2. Hence that information needs no be after how the pCO2 vertical profile is computed. Then it finishes off with a comment that temperature is partly driving the vertical gradient. However, Fig. 7 show that non-thermal drivers exert a greater control than temperature.

    **AR:** In section 3.9 we discuss the vertical gradient in $pCO_2$ or, more precisely, the difference between pCO2 at 0-to-4 m and $pCO_2$ at 8-to-12 m. To do that one uses $pCO_2$ calculated from pH and $A_T$ as $pCO_2$ is only measured in the Ferrybox (from water at 11 m depth). Unfortunately, there is no surface pH data between May and July. For clarity, the following sentence has been added to the legend of Fig. 9: *Data are missing in May to July because no surface pH data is available during this period.*

**RC:** Paragraph staring on line 240. I have difficulties with the signs here. First in the paragraph above it reads that the 11 m pCO2 overestimate the surface water values by 17 uatm (clear from Fig. 10), but in this paragraph it reads "correcting for the underestimation of 17 uatm ..". This in combination with the first presented air-sea flux of 20 mol/(m2yr) and the second -17 mol/(m2yr) does not make sense. To control if the pCO2 estimated from pH and salinity derived AT at 0-4 m depth is comparable to the measured it would be nice to see how the pCO2 estimated from pH and salinity derived AT at 11 m compare to the measured.

> **AR:** This paragraph is indeed inaccurate and confusing. An additional source of confusion is that the figures are not correctly located in the text. Additionally, David Ho also brought to our attention that it would be useful to bracket the air-sea $CO_2$ flux using the gas exchange parameterisation by wind speed designed in offshore settings that we had initially used and another parameterisation for wind-fetch-limited environments. The text has been extensively edited and the paragraphs now read:
>
> *For the 9 months when data are available, monthly median $pCO_2$ normalized at in situ temperature at 11 m vs 0-4 m are well correlated ($r^2 = 0.81$) but $pCO_2$ is higher at the surface than at 11 m, with a median difference of 17 μatm (Fig. 10).*
>
> *The air-sea $CO_2$ flux estimated from $pCO_2$ at 11 m is negative, indicating a $CO_2$ influx from the atmosphere, every month of a composite year (Fig. 11). The gas exchange coefficient k is notoriously difficult to measure. It is often parameterised by wind speed which is known to work well in deep waters offshore (Ho, 2006). In shallow areas, parameters other than wind speed become important. Dobashi and Ho (2023) proposed a formulation which might work better in wind-fetch-limited environments. Here we are bracketing the air-sea $CO_2$ flux using these two parameterisations. The annual air-sea flux ranges from -10.2 to -20.2 mol $CO_2$ $m^{-2}$ $yr^{-1}$, respectively with the formulations of Dobashi and Ho (2023) and Ho (2006). Correcting for the fact, discussed above, that surface $pCO_2$ is higher than $pCO_2$ at 11 m above leads to fluxes of -16.8 and -9 mol $CO_2$ $m^{-2}$ $yr^{-1}$ with the two parameterisations.*

**RC:** Abstract: Not all the parameters mentioned are determined every hour.

> **AR:** That is correct. The text has been revised accordingly.

**RC:** Line 5 of abstract. Specify that 11 m is the bottom/sampling depth.

> **AR:** Done.

**RC:** Line 19. Spell out what fastest and largest changes the Arctic Ocean exhibit.

> **AR:** This is spelled out in the subsequent sentences.

**RC:** Line 25-26. Delete the first "projected" in the text "The projected decrease in pH is projected to be larger in ....".

> **AR:** Done.

**RC:** Fig 1. Add (C) to legend and change a, b, c and d to 1, 2, 3 and 4.

> **AR:** Done.

**RC:** Line 63. Set the number of outliers in relation to the total number of determinations.

> **AR:** Good point. The sentence now reads *The number of outliers discarded was 38 and 41, respectively for $C_T$ and $A_T$ (out of a total number of samples of 229 and 236).*

**RC:** Table 1. Use the letter $\mu$ instead of mu for pCO2, as in the text.

**AR:** Done.

**RC:** Line 155. Fig. 3 should be Fig. 4.

**AR:** Changed, see below.

**RC:** Fig. 4. Wrong figure has been posted, is the same as Fig.3. Legend. One AT should be CT.

**AR:** Thanks for spotting that. These mistakes have been fixed.

**RC:** Table 3. Give information of what Q1 and Q3 stands for.

**AR:** Done.: *Q1 and Q3 are the first and third quartiles.*

**RC:** Line 2001. Add C after Fig. 5, and in next line insert 5 instead of ?? after Fig.

**AR:** Sone.

**RC:** Line 210 mention that numerous streams add freshwater in June to August. Another important source is melting sea ice and calving glaciers that add freshwater to the fjord system.

**AR:** Agreed. Text changed accordingly.

**RC:** In line 220 it reads that pH information should be given in Fig. 8, but it is not. It would have been nice to see that but if the data do not allow then delete pH here.

**AR:** Agreed, "pH" deleted.

**RC:** Fig 8. It states that the density gradient is given in C. But it cannot be, but maybe sigma. Please clarify.

**AR:** The legend is actually correct. Fig. 8C shows the difference in seawater **density** between surface and deep.

**RC:** Line 250. Change "Its it ranges.."

**AR:** Done.

---

## Author Comment (AC2)

**Reply to referee #2 of the manuscript "High-frequency, year-round time series of the carbonate chemistry in a high-Arctic fjord (Svalbard)"**

Jean-Pierre Gattuso et al.

June 5, 2023

We are very grateful to referee Yuanxu Dong for his/her constructive comments which greatly improved the manuscript. Below is a point-by-point reply (**RC**: referee comment; **AR**: author reply)

**RC:** This work provides valuable long-term carbonate observations and guidance on instrument setup, data processing and quality control for the coastal ocean measurements of carbonate species. The dataset would be a great contribution to the Arctic carbon cycle community and the method will be well-welcomed by the coastal ocean community. I think it is suitable to be published in ESSD upon resolving the minor corrections below. By the way, my expertise mainly allows me to comment on the pco2 and CO2 flux-related contents. Please refer to the other reviewer's comments for the remaining parts. Also, I am not a native English speaker, but the writing looks good to me.

    **AR:**

**General and specific comments:**
**RC:** Line 6: Is it possible to use the data present in this study to back-calculate the dissociation constants?

    **AR:** As far as we know, this is not possible..

**RC:** Line 7: '...remains unsettled for Arctic waters'. How representative the water at the measured location for the entire Artic waters?

    **AR:** The important point is cold and low salinity waters. "Arctic" has been replaced by "polar".

**RC:** Line 7: Does the stratification related to the ocean depth? If no, suggest removing 'despite the shallow depth'. Also, I did not see any discussion about this in the main text.

    **AR:** Stratification (or vertical gradient) is discussed in section

**RC:** Line 10: in the main text, the value is 17. Keep consistency.

    **AR:** Now the main text says 16.8.

**RC:** Line 12: 'are understood the least'. Why the least? Any reference for this? I was thinking the Antarctic and Southern Ocean are less understood because of the remote and limited measurements.

    **AR:** Agreed but we wrote "among". In any case, the sentence has been modified and now reads *Despite their major importance, Arctic shelves are among the coastal areas which are understood the least.*

**RC:** Line 20: I am curious why the Arctic SST increasing rate right now is not significantly higher than other regions considering the greatest future warming?

**AR:** This is indeed a very interesting question which goes beyond the scope of the present manuscript.

**RC:** Line 48-49: Fig. 1 A, B, C. The figure caption uses the capital A, B, C to represent the subplots. Keep consistency.

**AR:** Done.

**RC:** Line 50-51: I am wondering if this sulfuric acid will influence the pH and carbonate measurements. May quickly dilute by the water mixing?

**AR:** Wde have added explanation in the revised manuscript:  *To prevent biofouling of the sensors, every night at 00:10, a sulfuric acid (4% for 10 min) flush of the entire sensor system was followed by a rince with freshwater (30 min) prior to switching again to measuring mode. Data were not used for a total duration of 60 min after the initiation of the flush.*

**RC:** Figure 1: worth to check all the figure captions. Here a, b, c, d should be 1, 2, 3, 4 I think. In addition, add (C): Svalbard (A), Kongsfjorden and Ny-Ålesund (B), and (C) observational set-up. . .

**AR:** Corrected.

**RC:** Line 63: 'The number of outliers discarded was 38 and 41'. How many observations are in total?

**AR:** This information is provided in the revised version of the manuscript. It reads: *The number of outliers discarded was 38 and 41, respectively for $C_T$ and $A_T$ (out of a total number of samples of 229 and 236).*

**RC:** Line 98: consider removing '. . .' in the bracket.

**AR:** Done.

**RC:** Line 114: 'The gas exchange parameterization as a function of wind speed of Ho et al. (2006) was used'. I like Ho et al. (2006), but not sure if Ho et al. (2006) parametrization is the best here at a coastal environment. First, the (Ho et al., 2006) was derived from the open ocean (Southern Ocean) environment, while the (Nightingale et al., 2000) was derived from the coastal sea (the North Sea), which may be better here. Second, the K in the very coastal area (very shallow seawater) might be different from the K in a relatively open ocean (see (Yang et al., 2019)). May not need to change the K parameterisation here, but worth to add this information.

**AR:** The referee is correct. We have added the parameterization of Dobushi et al. (2023) which may be more applicable to shallow, wind-fetch-limited environments. The text has been extensively modified. See sections 2.7 and 3.9 as well as Fig. 11.

**RC:** Line 125: not clear to me which time-series data is mentioned here.

**AR:** We are merely saying that, according to the best of our knowledge, there has been until now high frequency, multi-year time-series.

**RC:** Figure 2: B: Monthly; C: Monthly

**AR:** Done.

**RC:** Figure 4: Here I did not see pco2 plot. Does the middle panel represent pco2? In addition, "aragonite $\Omega_a$ calculated from AT and AT", two AT, should be a typo.

    **AR:** Sorry, this was the wrong figure. Now the right figure, showing the $pCO_2$ panel, is included.

**RC:** Line 173-175: Looks to me that the performance of Papadimitriou et al. (2018) and Lueker et al. (2000) is similar. Worth to explain why 'the formulations of Papadimitriou et al. (2018) performs better on our data set'.

    **AR:** The referee is correct. The text has been revised as follows *In conclusion, the formulations of Lueker et al. (2000) and Papadimitriou et al. (2018)have similar performances with our dataset and generally perform better than those of Millero et al. (2002) and Sulpis et al. (2020). The formulation of Papadimitriou et al. (2018) is seldom used and the de facto standard has become the formulations of Lueker et al. (2000), which we have used in the present study.*

**RC:** Line 175: remove one right bracket.

    **AR:** Done.

**RC:** Line 195: 'The relationship between the measured and calculated pCO2 (blue line) is relatively poor'. What is the reason of this?

    **AR:** We are unable to pinpoint a single reason.

**RC:** Figure 5: add A after 5.

    **AR:** Done.

**RC:** Figure 6: explain the blue and red dots in subplots A-E. In addition, please check here 'In panels F-J, the orange lines [to add] indicate the medians, boxes show the first and third quartiles and the interquartile range'. Some typos.

    **AR:** Done.

**RC:** Line 222-223: 'Temperature is lower by up to 2 °C in the deep layer from January to October and higher by up to 0.3 °C in November and December.' Lower than the surface layer, higher than the surface layer?

    **AR:** Yes. Corrected accordingly.

**RC:** Figure 8: Any thoughts on why the salinity and temperature at surface is higher than at deeper water in December? Caption: A, B, C, not a, b, c.

    **AR:** Done.

**RC:** Line 232: Figure 5B shows that the calculated pco2 disagree with the directly measured pco2. Not sure how good the calculation here is. Consider providing the uncertainty of the calculated pco2.

    **AR:** According to the seacarb function "errors", the typical error in calculating $pCO_2$ from $A_T$ and $C_T$ is 14 $\mu$atm.

**RC:** Line 236 & Figure 10: here in the text, surface pco2 ¡ deep pco2, but figure 10 shows opposite. Please check and revise.

    **AR:**

**RC:** Line 239: It would be interesting to include some discussion about the surface pco2 ¿ deep pco2 scenario in December.

    **AR:**

**RC:** Line 240-242: Here is quite confusing. Does the value 20 missed a '-' (i.e., -20?)? 'Correcting for the underestimation of 17 μatm. . . ', I think it is overestimation? But if the uncorrected flux is -20 mol m−2 yr−1, after the 17 uatm correction, the flux should be higher than 20 in magnitude. Please double check and clarify carefully, because this is quite important for the conclusion.

    **AR:** The referee is correct: this sentence was confusing. The text now reads: *For the 9 months when data are available, monthly median pCO$_2$ normalized at in situ temperature at 11 m vs 0-4 m are well correlated ($r^2 = 0.81$) but pCO$_2$ is higher at the surface than at 11 m, with a median difference of 17 μatm (Fig.10).*

**RC:** Line 243-245: I guess here try to mention the entire Arctic Ocean carbon uptake. But I am wondering how representative for the Arctic Ocean this location is? I feel like this is just a coastal ocean, which is likely different from the relatively open seawater and also different from other coastal oceans

    **AR:** Agreed, this is why we compare with estimates for the coastal ocean from the literature.